# Film Dressings Based on Hydrogels: Simultaneous and Sustained-Release of Bioactive Compounds with Wound Healing Properties

**DOI:** 10.3390/pharmaceutics11090447

**Published:** 2019-09-02

**Authors:** Fabian Ávila-Salas, Adolfo Marican, Soledad Pinochet, Gustavo Carreño, Oscar Valdés, Bernardo Venegas, Wendy Donoso, Gustavo Cabrera-Barjas, Sekar Vijayakumar, Esteban F. Durán-Lara

**Affiliations:** 1Centro de Nanotecnología Aplicada, Facultad de Ciencias, Universidad Mayor, Huechuraba 8580000, Región Metropolitana, Chile; 2Instituto de Química de Recursos Naturales, Universidad de Talca, Talca 3460000, Maule, Chile; 3Vicerrectoría de Investigación y Postgrado, Universidad Católica del Maule, Talca 3460000, Maule, Chile; 4Department of Stomatology, Faculty of Health Sciences, University of Talca, Talca 3460000, Chile; 5Technological Development Unit (UDT), Universidad de Concepción, Av. Cordillera 2634, Parque Industrial Coronel, Coronel 4191996, Biobío, Chile; 6Nanobiosciences and Nanopharmacology Division, Biomaterials and Biotechnology in Animal Health Lab, Department of Animal Health and Management, Alagappa University, Science Campus 6th Floor, Karaikudi-630004, Tamil Nadu, India; 7Bio and NanoMaterials Lab, Drug Delivery and Controlled Release, Universidad de Talca, Talca 3460000, Maule, Chile; 8Departamento de Microbiología, Facultad de Ciencias de la Salud, Universidad de Talca, Talca 3460000, Maule, Chile

**Keywords:** sustained release, wound healing, crosslinking, allantoin, equilibrium swelling ratio, accumulative release, thermogravimetric analysis

## Abstract

This research proposes the rational modeling, synthesis and evaluation of film dressing hydrogels based on polyvinyl alcohol crosslinked with 20 different kinds of dicarboxylic acids. These formulations would allow the sustained release of simultaneous bioactive compounds including allantoin, resveratrol, dexpanthenol and caffeic acid as a multi-target therapy in wound healing. Interaction energy calculations and molecular dynamics simulation studies allowed evaluating the intermolecular affinity of the above bioactive compounds by hydrogels crosslinked with the different dicarboxylic acids. According to the computational results, the hydrogels crosslinked with succinic, aspartic, maleic and malic acids were selected as the best candidates to be synthesized and evaluated experimentally. These four crosslinked hydrogels were prepared and characterized by FTIR, mechanical properties, SEM and equilibrium swelling ratio. The sustained release of the bioactive compounds from the film dressing was investigated in vitro and in vivo. The in vitro results indicate a good release profile for all four analyzed bioactive compounds. More importantly, in vivo experiments suggest that prepared formulations could considerably accelerate the healing rate of artificial wounds in rats. The histological studies show that these formulations help to successfully reconstruct and thicken epidermis during 14 days of wound healing. Moreover, the four film dressings developed and exhibited excellent biocompatibility. In conclusion, the novel film dressings based on hydrogels rationally designed with combinatorial and sustained release therapy could have significant promise as dressing materials for skin wound healing.

## 1. Introduction

The primary function of the skin is to serve as a protective barrier against external hazards. The loss of integrity of large portions of the skin as a result of injury or disease may lead to a major disability or even death [1,2]. Therefore, wound healing is a fundamental physiological process that restores skin integrity, aiming to repair the damaged tissues [3]. Given its importance, the sequence of events of wound healing has been extensively studied for several decades [4]. The mechanism of wound healing is very complex, involving several physiological events such as coagulation, inflammation, cell proliferation, matrix repair, epithelization and remodeling of the scar tissue [5]. Interruption or deregulation of one or more phases of the wound healing process leads to non-healing (chronic) wounds [6].

Dressing materials, which are used for wounds or burns, are known as “artificial skin”. They should possess properties of normal skin to accelerate the recovery of wounded or destroyed skin areas. One of the most studied is the hydrogel dressing [7]. Hydrogels are 3D, hydrophilic and polymeric networks capable of absorbing large amounts of water or biological fluids [8], but do not dissolve when brought into contact with water [9]. Due to their high water content, porosity and soft consistency, they closely simulate natural living tissue, more so than any other class of synthetic biomaterials [8]. The network crosslinked by covalent bonds is classified as a chemical gel, while the formation of a physical gel takes place via a physical association between polymeric chains [10]. Compared with other biomaterials, hydrogels have the advantages of increased biocompatibility, tunable biodegradability, and porous structure, among others. However, owing to the low mechanical strength and fragile nature of the hydrogels, the feasibility of applying hydrogels is still limited. Thus, novel hydrogels with stronger and more stable properties are still needed and remain an important direction for research [11].

The exclusive physical properties of hydrogels have aroused particular interest in their use in drug release applications. Their highly porous structure can be easily tuned by controlling the crosslink density (cross-linking degrees) in the gel matrix and the affinity of the hydrogels for the aqueous medium in which they are swollen [12]. Their porosity also allows the loading of drugs into the gel matrix and subsequent drug release at a rate dependent on the diffusion coefficient of the small molecule or macromolecule through the gel network [13]. The properties that the drug delivery (usually governed by passive diffusion mechanisms) has also depend on factors such as hydrogel mesh sizes, stimuli-sensitivity and hydrogel capacity, among others [14].

Therefore, the structural properties of hydrogels and their affinity for certain bioactive molecules will depend directly on the selection of constituent polymers and the type of crosslinker that will form the polymeric crosslinked mesh. Among the polymers most used for the preparation of hydrogels for the treatment of wound healing are chitosan [15] and polyvinyl alcohol (PVA) [16].

For this study, PVA was selected because it is a biocompatible polymer and non-toxic to humans [17,18]. The formation of hydrogels from PVA can be performed by chemical methods, which involve the formation of interactions and bonds between the PVA chains and the functional groups of the crosslinking agents [19]. The concentration of crosslinker affects the porous structure, swelling features and mechanical strength. By setting the suitable degree of crosslinking, it is possible to prepare super-porous hydrogels with the desired characteristics. This will provide a platform to design novel drug delivery systems [20]. There are scientific studies showing that hydrogels based on PVA formulations cross-linked with specific crosslinkers are excellent bioactive compound releasing agents, especially at the dermal level: PVA-Glutaraldehyde [21], PVA-Ethylenglycol [22], PVA-Chitosan [23], PVA-Collagen [24], PVA-Cellulose [25], PVA-starch [7], PVA-Gelatin [26], PVA-TEOS [27], PVA-Heparin [28], PVA-poly(AAm) [29], among others.

PVA crosslinked with organic acids generates flexible and transparent hydrogels [18] capable of interacting with water-soluble compounds [30]. Its flexibility allows it to be easily handled during the treatment of wounds, providing a stronger mechanical protection. On the other hand, its transparency allows evaluating the process of healing step by step [18].

The organic acids or bifunctional molecules of interest in this study have two carboxylic acid functional groups at both ends of its structure (dicarboxylic acid molecules, for example, succinic acid), which through an esterification process can generate covalent bonds with the hydroxyl groups present in the polymer chains of PVA, generating the crosslinking and porosity of the structure [18].

The hydrogels from PVA must meet a number of requirements including biocompatibility, suitable porosity, swelling, mechanical strength and degradation properties. As mentioned earlier, all these properties are affected by the kind and concentration of polymers employed in hydrogels as well as by the cross-linking type and density [13,31]. Therefore, the porosity is directly related to the structure of the bifunctional molecule with low molecular weight, the length of its skeleton and the amount of ester bonds it can generate with the PVA chains. For example, the chemical hydrogel has been synthesized with specific dicarboxylic acids (DCA) [18,32].

As is well known, wound healing is a complex and sequenced process formed by several phases. In this context, the properties of hydrogels allow utilizing a delivery system of drug combinations simultaneously (multi-target therapy) [33]. This property of hydrogels could play a key role as a simultaneous delivery system of therapeutic agents for the wound healing process (constituted by several coordinated stages). Congruent with the above, drug combination therapy (directed at multiple therapeutic targets) improves treatment response and minimizes adverse events [34,35]. Due to their inherent properties, hydrogels are able to efficiently encapsulate and deliver in a controlled release manner [14]. Additionally, these materials must possess properties similar to normal skin: not possess toxins, provide an environment that prevents drying of the wound, reduce the penetration of bacteria, avoid losses of heat, water, proteins and red blood cells, in addition to promoting a rapid healing [36,37]. Thus, the use of biomaterials for the treatment of wounds is an area of interest for the scientific and medical community.

Therefore, the goal of the present article was to rationally develop hydrogel polymer formulations based on PVA crosslinked with a series of crosslinkers to improve the wound healing process of complex injuries through a simultaneous and sustained-release of allantoin, resveratrol, dexpanthenol and caffeic acid in the skin mouse model.

## 2. Materials and Methods

### 2.1. Theoretical Section

#### 2.1.1. Building Molecular Structures

The three-dimensional (3D) structures of allantoin (AL), resveratrol (RES), caffeic acid (CA), dexpanthenol (DEX), 20 different PVA hydrogel nanopores (PVAnp) (Appendix A) and PVA chain (of five monomers long) were designed and built through MarvinSketch software version 19.1.0, ChemAxon Ltd., Budapest, Hungary [38]. For all 3D structures, their protonation states at pH 7.0 were considered. Their geometries were optimized using Gaussian software version 16, revision A.03, Inc., Wallingford, CT, USA [39] at Density Functional Theory level using the B3LYP method and 6-311+G(d,p) as the selected basis set. In this study, the DCA selected by Marican et al. 2018 [18] were evaluated. 

#### 2.1.2. In-Silico Calculation of Interaction Energies

Interaction energies (ΔE) between the 20 nanopores and the compounds studied were calculated using a computational strategy implemented by Avila-Salas et al. 2012 [28] which couples a Monte Carlo conformational sampling [40] and ΔE calculations at the semi-empirical quantum mechanical (SQM) level [41]. With this methodology, it is possible to quickly evaluate the energy contribution of each component in the PVAnp-compound binding affinity. ΔE was calculated for molecule1-molecule2 complexes. In this case, molecule1 represents each one of the 20 PVAnp (Appendix A) and molecule2 represents the four compounds studied (AL), RES, CA and DEX). 

### 2.2. Experimental Section

#### 2.2.1. Materials

Polyvinyl alcohol (PVA) 30-60 KDa, succinic acid (SA), aspartic acid (AA) malic acid (MALI), maleic acid (MALE), NaHCO_3_, acetonitrile (HPLC grade), allantoin, dexphantenol, caffeic acid and resveratrol analytical standards were purchased from Sigma-Aldrich (St. Louis, MO, USA). HCl and methanol (HPLC grade) K_2_HPO_4_ and H_3_PO_4_ were purchased from Merck (Darmstadt, Germany). All solutions were prepared using MilliQ water. 

#### 2.2.2. Synthesis of Selected Hydrogels Based on PVA, Dicarboxylic Acids and Bioactive Compound Loading

For this study, twenty hydrogels based on PVA and dicarboxylic acids (PDCAH) were proposed. However, according to the theoretical analysis, four candidates present the best interactions with the bioactive compounds (BC) of interest. Therefore, four PDCAH with different crosslinkers were synthesized. The preparations of these platforms were performed through the esterification of PVA with DCA according to the method from Rodríguez Nuñez et al. 2019 with minor modifications [12]. Briefly, the reactions were performed by mixing an aqueous solution of PVA with an aqueous solution of a specific DCA (20 wt %) in presence of 1 × 10^−1^ mol·L^−1^ HCl (pH 1). After that, each reaction was carried out under reflux at 90 °C in a necked flask with magnetic agitation. After 3 h, each pre-hydrogel solution was poured into a new flask and a specific amount of BC (allantoin, dexpanthenol, caffeic acid, and resveratrol) was added for it encapsulation, as depicted in Table 1. Then, each solution was homogenized by stirring for 1 h and sonicated for 60 min until a homogenized solution was obtained. After that, each mixture solution was put in an oven at 45 °C overnight until the crosslinking was complete. Then, the PSAH, PAAH, PMALIH and PMALEH with encapsulated BC were washed several times with NaHCO_3_ for removing the excess acid. Finally, the hydrogels were lyophilized in order to obtain the xerogel. Lastly, each formulation obtained was termed as PSAH-BC, PAAH-BC, PMALIH-BC and PMALEH-BC, respectively.

#### 2.2.3. Equilibrium Swelling Ratio of PDCAH

The water uptake process was estimated by equilibrium swelling ratio (% ESR) at desired time intervals. Each xerogel film was immersed in phosphate buffer saline (PBS) (pH 7.4) and acetate buffer (pH 3.0) at 25 °C for 21 h until swelling equilibrium was attained. The weight of the wet sample [W_w_ (g)] was measured after carefully removing moisture on the surface with an absorbent paper. The weight of the dried sample [W_d_ (g)] was determined after the freeze-drying process of the hydrogel. The ESR of the hydrogel samples was calculated as follows (Equation (1)):(1)$$\mathrm{ESR}\;(\%)=\frac{W_{w}-W_{d}}{W_{d}}\;\times \;100\%$$

#### 2.2.4. Infrared Spectroscopy

Fourier-Transform Infrared (FT-IR) spectra of PSAH, PAAH, PMALIH and PMALEH were recorded on a Nicolet Nexus 470 spectrometer (Thermo Scientific, Waltham, MA, USA) within the 4000–400 cm^−1^ spectral intervals. All spectra were obtained in KBr pellets from an average of 32 scans with 4 cm^−1^ resolution.

#### 2.2.5. Mechanical analysis

Tensile tests were performed by means of a dynamometer model 4301, Instron (Canton, Oh, USA) equipped with a 5 kN load cell. The measurements were performed on dumbbell-shaped films. The width and the length of the investigated films were 5 mm and 30 mm, respectively, while the thickness of each film was measured at five random points using a micrometer and the result was expressed as the average value. All the measurements were carried out at 25 ± 2 °C and 50 ± 5% relative humidity at a crosshead rate of 5 mm·min^−1^. The reported data are the average values of five measurements. The obtained stress-strain curves were used to calculate tensile strength (σm, MPa), elongation at break (ε, %) and Young’s Modulus (E, MPa). 

#### 2.2.6. Scanning Electron Microscopy Analysis

The Scanning Electron Microscopy (SEM) analyses were performed for all four formulations. The films morphology was analyzed using a scanning electron microscope (JEOL-JSM 6380, Tokyo, Japan) operated at 15kV. Surface and side views of cryogenically fracture films were examined. All samples were sputtered with a gold layer, around 40 nm in thickness, previous to the analysis.

#### 2.2.7. Sustained Release Kinetics of BC from PDCAH-BC

The BC content of each supramolecular PDCAH (PDCAH-BC) is depicted in Table 1. Each PDCAH-BC with a weight of 400 mg was disposed into a 10 mL tube and 5 mL of PBS (pH 7.4) was poured over the formulation as a release medium. The tubes were transferred to an orbital shaker incubator water bath (Farazteb, Iran) at 33.5 °C ± 0.1 °C (Skin Temperature) and shaken at 35 ± 2 rpm. At specific time intervals, the PBS was removed and replaced with an equal volume of PBS in order to maintain sink conditions throughout the study. The samples of each supramolecular formulation were analyzed by a Perkin Elmer series 200 HPLC system (Norwalk, CT, USA) with a UV-Vis detector. An YWG C-8 (250 mm × 4.6 mm i.d. × 10 μm) column was used for the analysis of samples. 20 μL of eluent was injected into the HPLC. The mobile phase used consisted of 20 mM K_2_HPO_4_ (pH 6.0, H_3_PO_4_)/Methanol (90:10, *v/v*), in isocratic mode, at a flow rate of 1.0 mL·min^−1^. The samples were monitored at 210 nm (allantoin and dexpanthenol) and 300 nm (caffeic acid and resveratrol) by absorbance detection at 30 °C.

The release of each BC from each supramolecular formulation was determined by applying the amounts of released and loading BC to the following relationship (Equation (2)):(2)$$\mathrm{Cumulative}\;\mathrm{BC}\;\mathrm{release}\;\left(\%\right)\;=\mathrm{Cumulative}\;\mathrm{amount}\;\mathrm{of}\;\mathrm{BC}\;\mathrm{released}\;\times \frac{100}{\mathrm{Inicial}\;\mathrm{amount}\;\mathrm{of}\;\mathrm{BC}}$$

#### 2.2.8. Wound Healing Testing on Dermal Models of Rats

##### Animals and Maintenance Conditions

The experiments were carried out in adult Sprague Dawley rats of 150–200 grs obtained from the animal facility from the Universidad de Talca. All the animal care and experimental protocol were reviewed and approved by Comité Institucional de Ética, Cuidado y Uso de Animales de Laboratorio (CIECUAL) of the Universidad de Talca (Project identification code: 11170155; approval date of the committee: 17 December 2017). The animals were maintained in standard environmental conditions (22 ± 2 °C, relative humidity 70–80%, 12-h light cycle). The animals were weighed at the beginning and at the end of the experimental period. In addition, the intake of water and food was recorded. The rats were fed a standard diet manufactured by Champion (6.4% moisture, 3.6% lipids, 6.7% protein, 7.3 ashes, 3.6 fibers, 72.4% carbohydrates). The animals had free access to water and food; the bed was changed three times a week. Each cage had a record of changes in behavior or intake that was filled daily by the personnel in charge. 

##### Experimental Procedure

The animals were divided randomly into groups (5 animals per group). At the start of the surgical procedure, all animals were sedated with isofluorane and anesthetized with a mixture of ketamine (ketostop, DrangPharmainvetec S.A)/xylazine (Xylaret, Agroland) in a ratio of 3:1 (2.2 μL/g by weight). Once the anesthetized animals were in the surgical plane, the trichotomy of the inter scapular area was performed with a hair clipper (Oster gold) and the area washed with 0.25% chlorhexidine soap. Then, one skin segment in the area of the back between the scapulae was removed; surgery was performed with a special scalpel or punch. The diameter of the biopsy was around 1 cm. The excised wounds were covered with to-be-tested hydrogels (PDCAH-BC and controls, 1.2 cm × 1.2 cm) and affixed with an elastic adhesive bandage. Two groups control were used in this experiment, Madecassol*™* a commercial product and PSAH (film dressing without BC). The commercial product was daily applied until day 14. The total duration of each test was 14 days. On day 7, the PDCAH-BC and control (film dressing) were removed to analyze their adhesion and the film dressing was not reapplied. From days 7–14, the natural wound healing process was analyzed, protocol modified by Murakami et al., 2010 [42]. The wounds were examined and photographed for measurement of wound size reduction. These results were expressed in area and were represented by the closure of the wound. Differences in wound closure between controls and treatments were compared macroscopically. Upon completion of wound-healing experiments, the animals were sacrificed by excess diethyl ether on day 14 after the surgery. The rate of wound closure, which represents the percentage of wound reduction from the original wound size, was estimated utilizing the following formula (Equation (3)):(3)$$\mathrm{Wound}\;\mathrm{healing}\;\mathrm{reduction}\;(\%)=\frac{\mathrm{wound}\;\mathrm{area}\;\mathrm{day}\;0-\;\mathrm{wound}\;\mathrm{area}\;\mathrm{day}\;14}{\mathrm{wound}\;\mathrm{area}\;\mathrm{day}\;0}\;\times \;100$$

Values are expressed as a percentage of the healed wounds ± SD.

#### 2.2.9. Histological Analysis 

The histological analysis was oriented to the microscopic observation of the wound closure and was intended to compare the wound healing process. 5 μm thick sections from rat skin biopsies were used on silanized slides with 2% 3-aminopropyltriethoxysilane in acetone. The sample corresponded to rat skin affixed in 4% formaldehyde in 0.075 M sodium phosphate buffer pH 7.3, decalcified and embedded in paraffin. The sections were dewaxed and rehydrated following the routine protocol of the oral histopathology laboratory of the Universidad de Talca. The skin biopsies were stained with hematoxylin, eosin, Masson Trichrome and Giemsa.

#### 2.2.10. Cytotoxicity and Cell Viability

The cytotoxicity of PDCAH was evaluated on fibroblast cells. For this purpose, the viability of fibroblasts was assessed using MTT assay according to the protocol of Mossman et al. [43]. Briefly, the cells were seeded in 24-well plates (5 μL, 1.6 × 10^4^ cells per well) and 150 μL of Dulbecco’s Modified Eagle Medium (DMEM)-High medium was added and incubated for 24 h at 37 °C in 5% CO_2_. Then, the medium was substituted by 100 μL of fresh DMEM-High per well containing three different concentrations of PMALEH, PAAH, PSAH and PMALIH (500 μg·mL^−1^, 1500 μg·mL^−1^, and 2500 μg·mL^−1^ per formulation). Fresh medium without any PDCAH was used as a control. Cell viability was evaluated after 24 h by the MTT assays. Specifically, 5 μL of MTT solution (3 mg·mL^−1^ in PBS) and 50 µL of fresh medium were added to each sample and incubated for 4 h in the dark at 37 °C; formazan crystals were then dissolved in 100 µL dimethyl sulfoxide and incubated for 18 h. Supernatant optical density (o.d.) was evaluated at 570 nm (Spectrophotometer, Packard Bell, Meriden, CT, USA). Untreated cells were taken as control with 100% viability. The cell cytotoxicity of PDCAH was expressed as the relative viability (%), which correlates with the amount of liable cells compared with the negative cell control (100%). 

#### 2.2.11. Statistical Analysis

All experiments were performed in triplicate. Mean, standard deviation and Student’s t-test was performed to test the statistical significance in MTT assay studies and graphs were prepared by using Graphpad Prism 6. Statistical significance was set at *p* < 0.05.

## 3. Results and Discussion

### 3.1. In-Silico Interaction Energy Study

20 nanopores of the PVA hydrogels crosslinked with different dicarboxylic acids were designed (Appendix A). The interaction energy studies between the 20 nanopores and the compounds studied with healing activity were carried out: allantoin (AL), resveratrol (RES), caffeic acid (CA) and dexpanthenol (DEX). The results of these studies can be observed in Table 2.

According to the results obtained from ΔE, the pores generated between PVA and the succinic, malic, maleic and aspartic acids (marked in red in Table 2), have simultaneously better ΔE for the 4 compounds of interest. Therefore, hydrogels PVA cross-linked with these acids are good candidates to be evaluated experimentally.

### 3.2. Preparation of PDCAH

The preparation of formulations was performed as is depicted in Scheme 1 and Figure 1. Concisely, each hydrogel was prepared using polymerization by esterification in the presence of HCl as a catalyst. Once the pre-hydrogel was produced, the specific amount of each BC was added. With this methodology of loading, it is possible to obtain over 99% retention of the drug. The characterization analysis from FT-IR established the conjugation between PVA (–OH) and DCA (–COOH) into the PDCAH (The PDCAH characterization was performed without the loading drug (empty formulation). According to previous works, a crosslinking degree of 10:2 of PVA:DCA was prepared, which was kept constant due to its excellent features such as porosity, among others [12,18].

### 3.3. ESR Results

This characterization is very simple but very important at the same time since it confirms the hydrogel formation. In other words, if there is an increase in the swelling index, it means that the hydrogel matrix is absorbing the solvent and is not dissolved in the solvent; this being one of the most important features of a hydrogel [44]. In consequence, this characterization was made to confirm the preparation of the four hydrogel formulations with different crosslinker agents. Figure 2 shows the ESR for all four hydrogels. This figure displays an increase in the swelling index across time for all PDCAH. For all PDCAH, the swelling index in the first segment increased rapidly and afterwards slowly. This behavior may be due to the hydrogels reaching maximum constant swelling. The PSAH, PMALEH, PAAH and PMALIH reached the swelling equilibrium (zero order) at about 4–5 h. After 5 h, the PSAH, PMALEH, PMALIH and PAAH reached about 300, 400, 500 and 600% or swelling index, respectively, at pH 7.4. This may be due to the crosslinker agent polarity, which can be explained by the available hydrophilic groups in their structure that form hydrogen bonds with water molecules. In this context, we may conclude that the polarity led to an increase in ESR. Also, a significant difference for the set of formulations was observed between the two pH models. In all cases, the swelling index is higher a 7.4 than 3.0, observing a difference of ~80% of ESR between pH models. The data previously mentioned confirming that swelling behavior of the prepared hydrogels are pH-dependent owing to their ionic networks. In this sense, the four PDCAH absorbed a higher amount of water at 7.4 than 3.0. The ionic networks from the prepared hydrogels are provided by containing ionic pendant groups from the crosslinkers (SA, MALE, MALE, and AA), which have different types of pKa (depends on each crosslinker agent) [45]. This feature at a certain pH provides higher ionization degree in the hydrogel matrix, producing an intensification of electrostatic repulsion between chains from the networks. This electrostatic repulsion causes a higher uptake of solvent into the matrix, which increases the size of the hydrogel [12,18]. The swelling index observed could be related to the diffusion process where the encapsulated bioactive compounds could diffuse through swollen hydrogel networks toward the outside [46].

### 3.4. BC in Vitro Release Behavior of Supramolecular PDCAH

The encapsulation methodology carried out in this work was easier than conventional encapsulation by absorption. Specifically, the encapsulation was done through mixing BC with the pre-hydrogel solutions to encapsulate the BC into the PDCAH (PDCAH-BC), which decreased the encapsulation process time. On the other hand, this methodology allows for the loading of an exact amount of drugs (compared with the conventional method). The supramolecular PDCAH film was loaded with allantoin (5%), dexpanthenol (2%), caffeic acid (2%) and resveratrol (2%) according to the standard concentrations of bioactive compounds utilized in the dermatology area.

In order to analyze the in vitro release behavior of BC from PDCAH-BC, release profiles were obtained in physiological conditions (33.5 °C, PBS at pH 7.4). The samples were evaluated through HPLC method and the percent of the cumulative amount released was plotted over time. The BC cumulative release profiles are shown in Figure 3; the four PDCAH-BC provided a rapid release into the medium until 6 h, as shown for each BC and hydrogel. For example, 35% of allantoin, 38% of dexpanthenol, 52% of caffeic acid and 56% of resveratrol have been released from PMALEH-BC. After this initial fast release profile, PMALEH-BC showed a slower and steadier BC release into the medium for all cases.

The average release rate (%) of BC during the rapid phase (0 to 6 h) is depicted in Table 3 for each formulation. After 6 h, the rapid release rate changed toward a slower and sustained release. For PMALEH-BC, the average of the rapid-release phase was 0.58 mg/h, 0.63 mg/h, 0.87 mg/h and 0.93 mg/h for allantoin, dexpanthenol, caffeic acid and resveratrol, respectively. In contrast, the average of the slow-release phase was 0.11 mg/h, 0.11 mg/h, 0.08 mg/h and 0.08 mg/h for allantoin, dexpanthenol, caffeic acid, and resveratrol, respectively. In Table 4 and Table 5, all the average release values of each formulation and BC are provided.

The release patterns of the BC from the formulations were dependent on the crosslinker type. For instance, the higher release of allantoin and dexpanthenol occurred in PMALIH and PSAH. In contrast, the higher release of caffeic and resveratrol was produced in PMALEH and PAAH. Perhaps the crosslinker nature plays a key role in the release pattern of each biomolecule. Characteristics of crosslinkers, such as functional groups in its structure, polarity and ability to form hydrogen bonds, among others, affect the release patterns. 

### 3.5. FTIR Analysis

IR spectra of PDCAH using different dicarboxylic acids such as succinic, malic, aspartic, and maleic acid are presented in Appendix A, respectively. We can notice that all PDCAH spectra (PSAH, PMALIH, PAAH, and PMALEH) have most of PVA characteristic IR absorption bands (the PVA spectra, not shown here). In all spectra, we can find these representative bands that appear at around 3400 cm^−1^, between 2840 and 3000 cm^−1^, over 1688 cm^−1^. Signals between 1150 and 1085 cm^−1^ are attributed to a hydroxyl group (νOH), alkyl groups (νCH_2_), carbonyl groups (νC=O) and the ester group (C–O–C), respectively. The last two signals indicate that the crosslinking of PVA was due to the ester linkage formed between PVA and the different dicarboxylic acids used, as shown in Appendix A. Other important absorption bands are recorded in the PSAH, PMALIH, PAAH, and PMALEH spectra and prove the presence of the succinic, malic, aspartic, and maleic acids in their structures. For example, the peak at 1627 cm^−1^ present in the PMALEH spectra is a clear indication of the existence of –CO–CH=CH– stretching. On the other hand, in the PAAH spectra, we found two signals at 1630 and 1419 cm^−1^ characteristic to the CO–NH group from amide. In Appendix A (See Appendix A), we summarized the most characteristic bands with their assigned PVA and PVA cross-linked with succinic, malic, aspartic, and maleic acids. Finally, it is clear that the spectral changes obtained in the above analysis are evidence of cross-linking reactions between the hydroxyl group of PVA and the carboxylic groups of succinic, malic, aspartic, and maleic acids.

### 3.6. Mechanical Analysis

Mechanical properties of the PDCAH were summarized in Appendix A. Young modulus (E) specifies the stiffness or rigidity of the film; tensile strength (σ) indicates the tensile strength of the film up to breaking; and the elongation at break (ε) describes the flexibility or extensibility of the films up to breaking [47].

Mechanical analysis results showed that the highest tensile strength was obtained with PSAH (19.3 MPa), corresponding to PVA crosslinked with succinic acid, which was two times higher than PAAH, 1.5-times higher than PMALEH and 1.1 times higher than PMALIH. All these values are lower than what has been previously reported for pure PVA films [48], indicating that chemical crosslinking affects the mechanical properties. Based on these results, the changes in elongation at break and tensile strength appear to follow the same trend. Of the samples measured, sample two had the lowest recorded εB value (37.8%) followed by sample one (183.9%). The mechanical properties recorded in the case of sample one may be due to the film porosity observed by SEM analysis (Figure 4). By contrast, the change in the Young modulus followed a trend opposite to that of the other mechanical properties. The results indicate that the film from PAAH (176.4 MPa) possessed a rigid structure, low elasticity, and the lowest mechanical properties. This could be related to the highly fibrous and disorganized structure observed in SEM images (Figure 4) for this sample. A similar trend had been previously described for PVA films loaded with natural fibers [48]. From these results, it is clear that the nature of the crosslinker and possibly the degree of crosslinking could alter the mechanical properties of the films. 

### 3.7. SEM Analysis

In Figure 4, the results from the SEM analysis of PDCAH are shown. For PMALEH (see Figure 4a), a rough and porous surface is observed, which contrasts with the film inner view (see Figure 4b) that shows a non-porous and compact structure. In the case of PAAH and PMALIH (see Figure 4c,g), similar smooth surfaces are observed, but for sample PSAH, a rough and fibrous surface is displayed. On the other side, a highly fibrous and disorganized structure is observed at the bottom of the film side view from PAAH (see Figure 4d), changing to a more compact structure at the upper part. For samples PSAH and PMALIH (see Figure 4f,h), fibrous and compact structures are presented in the film side view, which is similar to that observed in PMALEH. It seems that film morphology would be highly influenced by the crosslinker chemical structure.

### 3.8. In Vivo Wound Healing Studies 

Figure 5 displays the images of the skin wound taken on day 0 and day 14, after treating with PAAH-BC, PSAH-BC, PMALIH-BC and PMALEH-BC and controls. After the 14th day, the PDCAH-BC treated wounds showed excellent results in all groups compared with the controls. The first characteristic detected was the growth of the new epidermis, which reduced towards the wound center in all treated wound lesions, resulting in a reduced area of the wounds. While the four formulations present good wound healing activity, there are some differences between them. For instance, PMALEH-BC achieved complete healing, while PAAH-BC, PSAH-BC and PMALEH-BC had a wound-healing ratio of 98 and 95%, 90% respectively. The wound healing process in the four proposed formulations was better than in the commercial and negative control (with a wound healing ratio of 85 and 40%, respectively). Such an excellent wound-healing effect of PDCAH-BC could be attributed to the synergistic effects among the bioactive compounds and their sustained release over the wound. On the one hand, allantoin has been reported to have numerous properties associated with wound healing, among them: hydrating and removing necrotic tissue, stimulating the cell mitosis as well as promoting epithelial stimulation, analgesic action and keratolytic activity [49]. On the other hand, the antioxidant agents such as resveratrol, dexpanthenol and caffeic acid have been reported with multiple activities, including anti-inflammatory and anti-bacterial activity [50,51,52,53,54]. Therefore, these compounds have shown positive effects for stimulating skin regeneration and promoting wound healing. Particularly, the treatment of the PDCAH-BC led to improved premature healing of the wounds. Hypothetically, the combinatorial therapy proposed in this work, where the bioactive compounds act on several therapeutic targets of the wound healing could be the key in the obtained results. 

### 3.9. Histological Analysis

Wound healing is a complex process which is involved of the following overlapping but well-defined stages: hemostasis, inflammation, migration, proliferation and remodeling [55,56]. Hematoxylin and eosin (H&E), Masson’s trichrome and Giemsa staining were utilized to evaluate the wound healing progress. The histology analysis of wounds covered with PDCAH-BC and controls on the 14th postoperative day is shown in Figure 6. In the Control a, the histological analysis shows a limited organization of the area under repair and the absence of reepithelialization was observed. On the other hand, no structural epithelium organization was observed and there was an absence of hair follicle formation. An increase in the cellular content such as fibroblasts, inflammatory and endothelial cells was observed, which is interpreted as a smaller organization of the connective tissue. A lot of blood vessels were observed, showing deep angiogenesis in the repair area. Moreover, moderate hemorrhage composed of extravasated red blood cells towards the deep area of the sample was observed. In addition, some incipient hair follicles at the edges of the healing area were detected (Figure 6a). In contrast, in sample c (PMALIH-BC), stratified epithelium and re-epithelialization in the repair zone was observed. The connective tissue presented a limited organization with high cellularity and vascularization. In sample d (PAAH-BC), the repair zone with a moderated organization and the connective tissue were detected. Moreover, high cellularity and vascularization were observed. In sample e (PSAH-BC), the repair zone with a moderated organization of connective tissue, vascularization and deep cellularity was revelated. The presence of hair follicles was evidenced in the deep area of scarring. 

In general, in all PDCAH-BC (Figure 6c–f) a better and faster reepithelialization process and retraction of the wound healing area was exhibited. A better organization of the granulation tissue in relation to control (a) was detected. Also, there was a greater delimitation of the scar area. In all PDCAH-BC, the epithelium showed signs of better structural organization, which included defined basal, spiny and superficial strata, as well as the beginning of granular stratum formation at the lateral edges of the repair area. The basal stratum of the epithelium had normotypic hyperchromatic cells compatible with proliferative activity in the reepithelialization zone. A similar characteristic in samples from PSAH-BC, PAAH-BC, PMALIH-BC and PMALEH-BC was observed. An apparent technical defect due to the detachment of scar tissue in samples d and e (defect during sample processing) was observed. However, the microscopical evidence suggests that the histological aspect of the wound healing process is better in sample from PMALEH-BC than PSAH-BC, PAAH-BC and PMALIH-BC as shown in Figure 6. In the case of the commercial product, similar features were detected than PMALEH, however, this formulation was daily applied until day 14 unlike the prepared formulations in this work applied only one time.

These results are in concordance with the closure evaluations (Figure 5) in which PMALEH-BC presented 100% wound closure. Moreover, these results could be supported by the release profile of PMALEH (Figure 3), in which the fastest release was produced with caffeic acid and resveratrol (with radical scavenging effects in hemostasis and inflammation [57]). Subsequently, PMALEH-BC allowed the slow release (later) of allantoin and dexpanthenol so that they could carry out their in later stages of wound healing (proliferation and remodeling [49,50]).

### 3.10. PDCAH Cytotoxicity Studies 

This study was performed to quantify the cytotoxicity of the prepared formulations on fibroblast cells. The cytotoxicity of the sterilized PMALEH, PAAH, PSAH and PMALIH was analyzed by a cell viability assay using L929 fibroblast cells after 24 h. Figure 7 displayed fibroblast cell viability exposed to three different concentrations of prepared formulations (a concentration range of 500–2500 μg·mL^−1^ for each PDCAH analyzed). As indicated in Figure 7, at 500 μg·mL^−1^ the cell viability is close to 100% for all four cases. On the other hand, it is observed that when significantly increasing PDCAH concentration, the fibroblast cell viability only declines slightly. In other words, among the concentration range of 1500 and 2500 μg·mL^−1^, the cell viability decreases between 95 and 88% for the all four prepared formulations, respectively. These results confirm that the PDCAH have minimum toxicity over the fibroblast cell model. These performed assays allow concluding that these proposed formulations could be biocompatible for medical applications. Therefore, PDCAH could be considered as safe formulations for sustained release of bioactive compounds with wound healing properties.

## 4. Conclusions

In this article, an *in-silico* strategy has been implemented to quickly evaluate the energy contribution of each compound (allantoin, dexpanthenol, caffeic acid and resveratrol) in the PVAnp-compound binding affinity. According to the results obtained from ΔE, the pores generated between PVA and the succinic, malic, maleic and aspartic acids have better ΔE for the 4 compounds of interest simultaneously. Therefore, hydrogels PVA cross-linked with these acids were the candidates to be evaluated experimentally.

Thus, based on the in-silico obtained results, novel film dressings based on hydrogels with unique properties were successfully prepared. Starting with the rational design of these formulations, this work concluded with in vivo wound healing studies that yielded promising results. Specifically, a series of hydrogels loaded with bioactive compounds with wound healing activity such as allantoin, dexpanthenol, caffeic acid, and resveratrol was developed. Moreover, an enhanced wound healing process in a full-thickness skin defect model with these formulations was demonstrated. The in vitro release studies exhibited that it is possible to carry out a combinatorial and coordinated sustained release of all four bioactive compounds, demonstrating an excellent strategy to achieve wound healing. These formulations have been designed by simply conjugating PVA chains and maleic, malic, aspartic and succinic acids as crosslinking agents.

The release profile of allantoin, dexpanthenol, caffeic acid and resveratrol exhibited some differences for each PDCAH. This difference seems to be governed by the affinity of the bioactive compound type and the crosslinking agent type (intermolecular interactions). The swelling index results concluded that these formulations based on hydrogels are stimuli-responsive to pH, time and crosslinking agent type. Moreover, in a great part of formulations, a rough, porous surface and good mechanical properties were observed. On the other hand, all four prepared film dressings showed good biocompatibility with L929 mouse connective tissue fibroblasts. The results revealed a viability of more than 88%.

In vivo studies, the PDCAH-BC treated wounds showed excellent results compared with the controls. Macroscopically, the growth of the new epidermis towards the wound center in all treated wound lesions was detected for all four cases (PMALIH-BC, PAAH-BC, PSAH-BC, and PMALEH-BC), resulting in a reduced area of the wounds. While the four formulations present good wound healing activity, there are some differences between them; this may be due to the nature of the crosslinking agent in each case. The microscopic evidence suggests that the histological aspect of the wound healing process is in concordance with the wound closure results. In conclusion, novel film dressings with simultaneous and sustained-release properties have been obtained and could be excellent candidates for skin wound healing.

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
