# Peer review of "Film Dressings Based on Hydrogels: Simultaneous and Sustained-Release of Bioactive Compounds with Wound Healing Properties"

_pharmaceutics, 2019, doi:10.3390/pharmaceutics11090447_

Round 1

Reviewer 1 Report

This paper is a very interesting and well documented study.

Please make the minor revison / suggested corrections as follows:

Line 342 – please write Figure 3 instead Figure 8

Please add the SD values for rapid-release phase of BC (%) until 6 h – see Table 3

The same comments for Tables 4 and 5.

Author Response

Dear reviewers and editor,

- The changes associated with the comments and suggestions of reviewer 1 were marked in yellow in the manuscript.

- The changes associated with the comments and suggestions of reviewer 2 were marked in green in the manuscript.

- The changes associated with the comments and suggestions of reviewer 3 were marked in light blue in the manuscript.

- Other changes made for the authors were marked in fuchsia in the manuscript.

Reviewer 2 Report

In this manuscript, authors have shown the hydrogel polymer formulations based on PVA crosslinked with a series of crosslinkers to improve the wound healing process of complex injuries through a simultaneous and sustained-release of allantoin, resveratrol, dexpanthenol and caffeic acid in the skin mouse model. Overall this manuscript is well written and straight forward from the original hypothesis to conclusion. I have several points to comment.

Overall, more in vivo experimental results are required to show the effect of healing. Immunofluorescence staining images related with involucrin, laminin, microvessels might be helpful for reader to understand the level of healing.

Once again, wound closing ratio should be included. Detailed explanation about the roles of bioactive compound in wound healing should be added in discussion part.

Is there any reason that acetate buffer was used for equilibrium swelling ratio analysis? Detailed explanation with appropriate reference is required.

Cell seeding method suggested in section 2.1.0 is not clear. Line 3 and following sentence is difference from each other.

Author Response

_____________ Reviewer 2 _____________

In this manuscript, authors have shown the hydrogel polymer formulations based on PVA crosslinked with a series of crosslinkers to improve the wound healing process of complex injuries through a simultaneous and sustained-release of allantoin, resveratrol, dexpanthenol and caffeic acid in the skin mouse model. Overall this manuscript is well written and straight forward from the original hypothesis to conclusion. I have several points to comment.

We thank the reviewer for their appreciation and comments.

(1) Overall, more in vivo experimental results are required to show the effect of healing. Immunofluorescence staining images related with involucrin, laminin, microvessels might be helpful for reader to understand the level of healing.

Response: We thank the reviewer for this comment and it will be considered for the next article in progress. However, for now, it would be unviable to carry out this type of study for time and cost reasons.

(2) Once again, wound closing ratio should be included. Detailed explanation about the roles of bioactive compound in wound healing should be added in discussion part.

Response: The detailed explanation about the roles of bioactive compounds in wound healing was added in the manuscript on page 15, lines 435-440. In addition, references 54 to 57 were added in the references section on the page 21, lines 689-697.

(3) Is there any reason that acetate buffer was used for equilibrium swelling ratio analysis? Detailed explanation with appropriate reference is required.

Response: The acetate buffer was used in this study only to evaluate the pH-dependent swelling of the film dressings at pH 3.0. In the text of the manuscript submitted, this behavior has been explained and supported with the respective references (page 10, line 321-332)

(4) Cell seeding method suggested in section 2.1.0 is not clear. Line 3 and following sentence is difference from each other.

Response: In line 261, the sentence “Succinctly, the cells were seeded in 24-well plates (1.6 x 104 cells per well). Then, 5 μL of cells and 150 μL of Dulbecco's Modified Eagle Medium (DMEM)-High medium were added and incubated for 24 h at 37 °C in 5% CO2” was modified for “Briefly, the cells were seeded in 24-well plates (5 μL, 1.6 x 104 cells per well) and 150 μL of Dulbecco's Modified Eagle Medium (DMEM)-High medium was added and incubated for 24 h at 37 °C in 5% CO2” on page 7, lines 261-263.

Reviewer 3 Report

This article entitled “Film dressings based on hydrogels: simultaneous and sustained-release properties of bioactive compounds with wound healing properties” by Fabián Avila-Salas et al proposes the rational modeling, synthesis and evaluation of film dressing hydrogels based on polyvinyl alcohol crosslinked with 20 different kinds of dicarboxylic acids.

Four crosslinked hydrogels were prepared and characterized by FTIR, mechanical properties, SEM and equilibrium swelling ratio. The in vitro results indicate a good release profile for all four analyzed bioactive compounds. In vivo experiments suggest that prepared formulations could considerably accelerate the healing rate of artificial wounds in rats. The histological studies show that these formulations help to successfully reconstruct and thicken epidermis during 14 days of wound healing.

The work is interesting and the working amount is essentially large. It is publishable if the authors take a few minor revisions in the following aspect.

1). In Scheme 1 and experimental section, the reflux was stated as 90-100 oC. In organic synthesis, temperature is critical and should be accurate. Under 90 oC or 100 oC could influence outcomes largely. Please specify the reaction condition with accurate temperature for each reaction of PSAH, PMALIH, PAAH, PMALEH.

2). Figure 4, scale bars should be placed bottom left, instead of in the middle.

3). Please provide statistical analysis for data in Figure 7 to see whether they are different or not. In section 2.2.10. Statistical analysis “These results were expressed as the means ± Standard Deviation (S.D.) from three replicates in order to minimize the experimental error.” This is just how you calculated the average value, it’s not statistical analysis.

Author Response

_____________ Reviewer 3 _____________

This article entitled “Film dressings based on hydrogels: simultaneous and sustained-release properties of bioactive compounds with wound healing properties” by Fabián Avila-Salas et al proposes the rational modeling, synthesis and evaluation of film dressing hydrogels based on polyvinyl alcohol crosslinked with 20 different kinds of dicarboxylic acids.

Four crosslinked hydrogels were prepared and characterized by FTIR, mechanical properties, SEM and equilibrium swelling ratio. The in vitro results indicate a good release profile for all four analyzed bioactive compounds. In vivo experiments suggest that prepared formulations could considerably accelerate the healing rate of artificial wounds in rats. The histological studies show that these formulations help to successfully reconstruct and thicken epidermis during 14 days of wound healing.

The work is interesting and the working amount is essentially large. It is publishable if the authors take a few minor revisions in the following aspect:

We thank the reviewer for their appreciation and comments.

(1) In Scheme 1 and experimental section, the reflux was stated as 90-100 ºC. In organic synthesis, temperature is critical and should be accurate. Under 90 ºC or 100 ºC could influence outcomes largely. Please specify the reaction condition with accurate temperature for each reaction of PSAH, PMALIH, PAAH, PMALEH.

Response: The temperature of reaction was corrected on page 4 line 161 and in scheme 1 on page 9.

(2) Figure 4, scale bars should be placed bottom left, instead of in the middle.

Response: The scale bar was placed in the bottom left. It is located in the manuscript on page 14.

(3) Please provide statistical analysis for data in Figure 7 to see whether they are different or not. In section 2.2.10. Statistical analysis “These results were expressed as the means ± Standard Deviation (S.D.) from three replicates in order to minimize the experimental error.” This is just how you calculated the average value, it’s not statistical analysis.

Response: As per the reviewer’s suggestion, we provided statistical analysis for the viability assay. Specifically, Student’s t-test was performed to test the statistical significance in MTT assay studies. The information was added in the manuscript on page 7, lines 275-277 (methodology) and in figure 7 on page 17.

Round 2

Reviewer 1 Report

The authors answered to all comments addressed by the reviewers.

Reviewer 2 Report

Authors have changed their manuscript according to my previous comments. I have no further suggestions.

Reviewer 3 Report

The authors addressed my comments well.